# Polyketide Derivatives from the Endophytic Fungus *Phaeosphaeria* sp. LF5 Isolated from *Huperzia serrata* and Their Acetylcholinesterase Inhibitory Activities

**DOI:** 10.3390/jof8030232

**Published:** 2022-02-26

**Authors:** Yiwen Xiao, Weizhong Liang, Zhibin Zhang, Ya Wang, Shanshan Zhang, Jiantao Liu, Jun Chang, Changjiu Ji, Du Zhu

**Affiliations:** 1Key Laboratory of Protection and Utilization of Subtropic Plant Resources of Jiangxi Province, College of Life Sciences, Jiangxi Normal University, Nanchang 330022, China; xyw1152858687@163.com (Y.X.); zzbbio@jxnu.edu.cn (Z.Z.); 2Key Laboratory of Bioprocess Engineering of Jiangxi Province, College of Life Sciences, Jiangxi Science and Technology Normal University, Nanchang 330013, China; lwz271508312@163.com (W.L.); jxwangya@126.com (Y.W.); zss18135437140@163.com (S.Z.); liujt@jxstnu.edu.cn (J.L.); changjun@jxstnu.edu.cn (J.C.); 3College of Chemistry and Biological Engineering, Yichun University, Yichun 336000, China; jichjiu999@jxycu.edu.cn

**Keywords:** *Phaeosphaeria* sp., secondary metabolite, polyketide, AChE inhibitor, biosynthetic pathways

## Abstract

The secondary metabolites of *Phaeosphaeria* sp. LF5, an endophytic fungus with acetylcholinesterase (AChE) inhibitory activity isolated from *Huperzia serrata*, were investigated. Their structures and absolute configurations were elucidated by means of extensive spectroscopic data, including one- and two-dimensional nuclear magnetic resonance (NMR), high-resolution electrospray ionization mass spectrometry (HR-ESI-MS) analyses, and calculations of electronic circular dichroism (ECD). A chemical study on the solid-cultured fungus LF5 resulted in 11 polyketide derivatives, which included three previously undescribed derivatives: aspilactonol I (**4**), 2-(1-hydroxyethyl)-6-methylisonicotinic acid (**7**), and 6,8-dihydroxy-3-(1′*R*, 2′*R*-dihydroxypropyl)-isocoumarin (**9**), and two new natural-source-derived aspilactonols (G, H) (**2**, **3**). Moreover, the absolute configuration of de-*O*-methyldiaporthin (**11**) was identified for the first time. Compounds **4** and **11** exhibited inhibitory activity against AChE with half maximal inhibitory concentration (IC_50_) values of 6.26 and 21.18 µM, respectively. Aspilactonol I (**4**) is the first reported furanone AChE inhibitor (AChEI). The results indicated that *Phaeosphaeria* is a good source of polyketide derivatives. This study identified intriguing lead compounds for further research and development of new AChEIs.

## 1. Introduction

Natural products are important sources of active compounds and play important roles in modern drug research and development. Fungi are considered an important group of microorganisms in the production of antitumor, immunosuppressant, antibiotic, antifungal, antiparasitic, anti-inflammatory, enzyme-inhibiting, and other active secondary metabolites [1,2]. Endophytic fungi reside in the internal tissues of living plants without causing apparent disease. Due to their unique ecological niche, endophytic fungi have become important sources of natural products to be screened for with unique chemical structures and biological activity [3,4]. Therefore, the natural product screening of endophytic fungi is currently a hot research topic [5,6,7,8,9]. In this sense, it is worth undertaking a constant search for novel compounds from endophytic fungal sources and paying attention to discovering potential drug candidates.

*Huperzia serrata* is a member of the *Lycophyllaceae* family called shezucao in China [10]. Huperzine A (HupA) was first isolated from *H. serrata* in the 1980s and was approved in the 1990s in China as an acetylcholinesterase inhibitor (AChEI) to treat Alzheimer’s disease (AD). These promising studies showed that endophytic fungi of *H. serrata* can synthesize HupA and similar compounds in host plants and also contains many novel compounds [10]. Thus far, many studies have been conducted on the diversity of endophytic fungi of *H. serrata*, but there are few studies on the isolation and screening of AChEIs in these endophytic fungi, and therefore further studies are needed. In our previous study, a total of 22 endophytic fungal strains showed strong AChE inhibitory activity (≥50%) [11]. As part of our ongoing research, we are currently characterizing the bioactive secondary metabolites of these endophytic fungi.

*Phaeosphaeria* sp. LF5 is an endophytic fungus isolated from the leaves of *H. serrata* [12]. Members of *Phaeosphaeria* have afforded a variety of natural products, such as polyketides, peptides, and terpenes [13,14]. Herein, *Phaeosphaeria* sp. LF5 was selected for screening for new AChEI natural products. We refermented the strain in solid substrate fermentation medium and then isolated 11 polyketide derivatives, which included three new compounds, aspilactonol I (**4**), 2-(1-hydroxyethyl)-6-methylisonicotinic acid (**7**), and 6,8-dihydroxy-3-(1′*R*, 2′*R*-dihydroxypropyl)-isocoumarin (**9**), and two new natural source-derived aspilactonols (G, H) (**2**, **3**). We also identified the absolute configuration of de-*O*-methyldiaporthin (**11**) for the first time (Figure 1). In addition, we detected their AChE inhibitory activity. Herein, the isolation, structural elucidation, and bioactivities of these isolated compounds are described.

Generally, furanone derivatives are polyketide metabolites found in *Aspergillus* [15,16]. They are classified into three structural types: linear (aspinonene), δ-lactones (aspinrone), and γ-lactones (isoaspinonene and aspilactonols) [17]. To date, it has been a challenging task to assign the absolute configurations of furanone derivatives due to the flexibility of their aliphatic sidechain in the partial polyketide unit [18,19]. Isocoumarins comprise a six-membered oxygen heterocycle (α-pyranone) with one benzene ring. Isocoumarins represent a group of natural compounds rich in lactones, which are mainly derived from the fungal polyketone pathway. These compounds have exhibited a wide range of biological functions, including antifungal, anti-inflammatory, insecticidal, and hepatoprotective activities [20]. However, the determination of their absolute configuration becomes quite challenging due to the high degree of free rotation of the steric centers on the chain, with the side chains connected to the nuclei of isocoumarin derivatives [21]. In the present study, the structures and absolute configurations of polyketide derivatives isolated from *Phaeosphaeria* sp. LF5 were elucidated by means of extensive spectroscopic data, including one- and two-dimensional nuclear magnetic resonance (NMR) spectrometry, high-resolution electrospray ionization mass spectrometry (HR-ESI-MS) analyses, and calculations of electronic circular dichroism (ECD).

## 2. Materials and Methods

### 2.1. General Experimental Procedures

Optical rotation values were determined on a JASCO P-1010 polarimeter (Jasco, Tokyo, Japan). UV spectra were recorded on a PerkinElmer Lambda 365 UV-Vis spectrophotometer (PerkinElmer, Hopkinton, MA, USA). High-resolution electrospray ionization mass spectrometry (HR-ESI-MS) data were measured on a Waters ACQUITY UPLC H-Class Q-TOF LC-MS spectrometer (Waters, Milford, MA, USA). High-performance liquid chromatography (HPLC) analysis was carried out on an ACQUITY UPLC H-Class System (quaternary solvent manager, sample manager, PDA detector, and ELS detector) using a YMC ODS (4.6 × 250 mm, 5 µm, 1 mL/min) column. MPLC was performed on a PuriFlash450 (Interchim, Los Angeles, CA, USA) with a Flash C18 cartridge (50 µm, 40 g, YMC, Kyoto, Japan). Semipreparative HPLC was performed on a Waters 2535 Quaternary gradient module with a FlexInject, 2489 UV–VIS detector and Fraction Collector Ⅲ (Waters, Milford, MA, USA). The NMR spectra were recorded on a Bruker Avance 400 MHz spectrometer using tetramethylsilane as the internal standard (Bruker, Ettlingen, Germany). Thin-layer chromatography (TLC) analyses were performed on glass precoated with silica gel GF254 glass plates. All reagents for the analysis were purchased from Xilong Scientific Co., Ltd. (Guangdong, China).

### 2.2. Fungal Material

The endophytic fungus *Phaeosphaeria* sp. LF5 was isolated from the leaves of *H. serrata* at the Chinese Academy of Sciences’ Lushan Botanical Garden in Jiangxi Province, China [12]. This strain was deposited in the culture collection of the Key Laboratory of Protection and Utilization of Subtropical Plant Resources of Jiangxi Province, Jiangxi Normal University.

### 2.3. Fermentation and Extraction

The endophytic fungus LF5 was cultivated in 100 Erlenmeyer flasks (1000 mL); each flask contained 80 g of rice and 120 g of H_2_O to create solid rice medium. The flasks were then static incubated at 28 °C for 40 days.

After the mycelia entered the static growth state, the rice solid fermentation was taken out of the Erlenmeyer flask, dried at 45 °C to remove the water, crushed before adding 80% ethanol, and ultrasonically agitated for 1 h. The static precipitation was filtered, and the above steps were repeated four times to obtain an ethanol extract. The filtrate was removed with a rotary evaporator (35 °C, 160 rpm). The ethanol crude extract was placed in 2000 mL of water and transferred to a separatory funnel. In turn, petroleum ether (PE), ethyl acetate (EA), and water-saturated butanol were used for extraction four times and were concentrated in vacuo to yield the combined crude extracts, PE extract (16.2 g), EA extract (76 g), n-butanol extract (105 g), and water extract (450 g).

### 2.4. Isolation and Purification

The EA extract (76.0 g) was dried and subjected to column chromatography on 200–300 mesh silica gel with different solvents of increasing polarity from PE to EA to MeOH to obtain eight fractions (Frs. 1–8) on the basis of TLC analysis.

Fraction 6 was purified by Sephadex LH-20 (GE Healthcare, Pittsburgh, PA, USA) (MeOH) to obtain three subfractions: Fr. 6.1–Fr. 6.3. Fraction 6.1 was further purified by semipreparative HPLC (CH_3_OH/H_2_O, 30:70, *v/v*) to yield **1** (12 mg, *t*_R_ = 4.0 min), **2** (9.5 mg, *t*_R_ = 2.9 min), and **3** (7 mg, *t*_R_ = 7.0 min). Fraction 8 was loaded onto a Sephadex LH-20 column and eluted with EA/CH_3_OH (8:2) to yield two subfractions: Fr. 8.1 and Fr. 8.2. Subfraction Fr. 8.2 was separated by semi-preparative RP-HPLC (CH_3_OH/H_2_O, 10:90) to generate **4** (6.8 mg, *t*_R_ = 11.0 min) and **5** (4.0 mg, *t*_R_ = 8.0 min). Fraction 3 was separated by Sephadex LH-20 using CH_3_OH as the eluting solvent and was then further purified via semipreparative HPLC CH_3_OH/H_2_O (30:70, *v/v*) to obtain compound **6** (2.9 mg, *t*_R_ = 6.0 min). Fraction 7 was further purified using Sephadex LH-20 (EA/CH_3_OH, 90:10, *v/v*) to yield two subfractions: Frs. 7.1 and 7.2. Subfraction Fr. 7.1 was separated by semipreparative reversed-phase HPLC (RP-HPLC) (CH_3_OH-H_2_O, 5:95) to produce **7** (4.5 mg, *t*_R_ = 4.5 min). Fraction 2 (PE/EA 7:3) was separated into three subfractions (Fr. 2.1–2.3) with Sephadex LH-20 using MeOH as a mobile phase. Subfraction Fr. 2.1 was further purified via semipreparative HPLC using CH_3_OH:H_2_O (30:70) as a mobile phase at a flow rate of 5 mL/min to yield **8** (10 mg, *t*_R_ = 36.0 min). Subfraction Fr. 6.3 was separated by preparative HPLC (CH_3_OH/H_2_O, 30:70, *v/v*) to yield compounds **9** (8 mg, *t*_R_ = 17 min), **10** (5 mg, *t*_R_ = 30.0 min), and **11** (4 mg, *t*_R_ = 45.0 min).

### 2.5. Acetylcholinesterase Inhibitory Activity In Vitro Assay

The determination of the in vitro AChE inhibitory activity of the endophytic fungal extracts and compounds **1**–**11** was performed according to the spectrophotometry method developed by Ellman et al. [22] and modified by Ortiz et al. [23]. Rivastigmine and HupA, two known AChEIs, were used as positive controls. The assay was carried out in a 96-well microtiter plate reader. In brief, a preincubation solution of 250 μL of phosphate buffer (200 mM, pH 7.7) that contained 15 μL of purified compounds/HupA, 80 μL of DTNB (3.96 mg of DTNB and 1.5 mg of sodium bicarbonate dissolved in 10 mL of phosphate buffer, pH 7.7), and 10 μL of AChE was prepared. The mixture was incubated for 5 min at 25 °C. After preincubation, 15 μL of the substrate AChI (10.85 mg in 5 mL of phosphate buffer) was added and incubated again for 5 min. The color developed was measured in a microwell plate reader at 412 nm (Molecular Devices, SpectraMax M2, San Jose, CA, USA). Percent inhibition was calculated through the following formula: (control absorbance–sample absorbance)/control absorbance × 100. The IC_50_ values were the means ± SD of three determinations.

### 2.6. ECD Calculations

In general, conformational analyses were performed by random searching in Sybyl-X 2.0 using the MMFF94S force field with an energy cut-off of 5 kcal/mol (Sybyl Software, version X 2.0, 2013) [24]. The results showed the five lowest-energy conformers for compounds **4**, **7**, **9**, and **11**. Subsequently, geometric optimizations and frequency analyses were implemented at the B3LYP-D3(BJ)/6-31G* level in PCM MeOH using ORCA4.2.1 [25,26]. All conformers used for property calculations in this study were characterized as stable points on a potential energy surface with no imaginary frequencies. The excitation energies, oscillator strengths, and rotational strengths (velocity) of the first 60 excited states were calculated by the time-dependent density-functional theory (TD-DFT) at the PBE0/def2-TZVP level in MeOH. The ECD spectra were simulated by the overlapping Gaussian function (half the bandwidth at 1/e peak height, sigma = 0.30 for all) [27]. The Gibbs free energies for the conformers were determined by using thermal correction at the B3LYP-D3(BJ)/6-31G** level, and electronic energies were evaluated at the wB97M-V/def2-TZVP level in PCM MeOH using ORCA4.2.1 [25,26]. To obtain the final spectra, we used the Boltzmann distribution theory and the conformers’ relative Gibbs free energy (∆G) to average the simulated spectra. The absolute configuration of the only chiral center was determined by comparing the experimental spectra to the calculated molecular models.

## 3. Results and Discussion

### Structure Elucidation

Compound **4**, a white powder soluble in methanol (MeOH), exhibited a pseudomolecular ion peak at *m/z* 187.0965 [M + H]^+^ (calculated for C_9_H_15_O_4_^+^: 187.0926) in the HR-ESI-MS spectrum, indicating a molecular formula of C_9_H_14_O_4_, and two degrees of unsaturation. The ^1^H NMR data indicated two methyls at *δ*_H_ 1.20 (3H, d, *J* = 6.1 Hz, H-10) and 1.39 (3H, d, *J* = 6.8 Hz, H-6); one olefinic methane at *δ*_H_ 7.36 (1H, br s, H-4); and three oxymethines at *δ*_H_ 3.58 (1H, m, H-9), 3.60 (1H, m, H-8), and 5.09 (1H, br q, *J* = 6.8 Hz, H-5). The ^13^C-NMR spectra revealed one ester carbonyl (*δ*_C_ 176.5), one olefinic methine (*δ*_C_ 154.2), one nonprotonated sp^2^ carbon (*δ*_C_ 131.7), three oxygenated methines (*δ*_C_ 71.6, 74.8, and 79.8), one methylene (*δ*_C_ 29.6), and two methyls (*δ*_C_ 18.9 and 19.1).

The HMBC spectrum showed correlations between H-10 (*δ*_H_ 1.20)/C-8 (*δ*_C_ 74.8) and H-10 (*δ*_H_ 1.20)/C-9 (*δ*_C_ 71.6), as well as H-9 (*δ*_H_ 3.58)/C-7 (*δ*_C_ 29.6), H-9 (*δ*_H_ 3.58)/C-8, and H-9 (*δ*_H_ 3.58)/C-10. C-8 was correlated with C-3, C-7, and C-9 (Figure 2). When combined with the peak shape analysis of H-10 (*δ*_H_ 1.20, d, *J* = 6.1), C-9 was found to be connected to C-10, and C-8 was connected to C-9. Since C-8 and C-9 are methylene carbons and there is no nitrogen in the molecular formula, when combined with the chemical shift value, C-8 and C-9 were found to be connected to hydroxyl groups. H-7 is related to C-8, C-9, C-3, and C-4, and H-7 is methylene, but there were two groups of different H signals. Therefore, it can be inferred that one side of C-7 was connected to C-8. C-3 had the same characteristic signals as H-4, H-7, and H-8. The carbon shift signals of C-3 and C-4 were *δ*_C_ 131.7 and *δ*_C_ 154.2, respectively. HSQC indicated that C-4 was connected by protons, and it can be concluded that C-3 and C-4 were connected by a double bond and that C-4 was a quaternary carbon with two substitutions, one of which was connected with C-7. According to the HMBC cross-peak correlation of H-4/C-2 (*δ*_C_ 176.5), H-7/C-2, and C-3, we were able to infer that the side-chain fragment was attached to α,β-unsaturated-γ-lactone. HSQC indicated that C-2 was not connected to a proton. When combined with the molecular formula, C-4 was found to be a carbonyl group, H-4 was related to C-5, H-5 was related to C-6, and H-4 was related to HMBC. It was concluded that C-5 was connected to C-4, C-6 was connected to C-5, and the chemical shift value of C-5 was *δ*_C_ 79.8. When combined with the molecular formula, C-5 was also found to be connected to oxygen. Since the unsaturation degree of the compound was 2, the double bond between C-4 and C-3 occupied an unsaturation. An unsaturation remained, and there was no other double-bond carbon signal in the compound: hence, it is inferred that there was a cyclization system in which the chemical shift of C-2 was lower-field than that of the conventional carbonyl group. It was inferred that the other side of C-2 was connected to the oxygen and that C-5 and C-2 passed through the oxygen to form a lactone ring. The ^1^H (CD_3_OD, 400 MHz) and ^13^C-NMR (CD_3_OD, 100 MHz) data are listed in Table 1.

The ^13^C chemical shift calculation was carried out at the B3LYP-D3(BJ)/6-31G** level to obtain the accurate relative configuration of **4**. In addition, the absolute configurations of **4** (5*R*, 8*R*, and 9*S*) were established by comparing electronic circular dichroism (ECD) calculations at the PBE0/def2-TZVP level with the experimental one (Figure 3). In addition, high correlation coefficients (*R*^2^) between experimental and calculated chemical shifts were shown, with 0.9985 for **4** (Figure 4), indicating that the *δ*_C_ of **4** matched the calculated *δ*_C_ very well, which confirmed the framework of **4**. The structure of compound **4** was determined to be (*R*)-5-((8*R*, 9*S*)-8, 9-dihydroxybutyl)-5-methylfuran-2(5H)-one, so compound **4** was named aspilactonol I, as shown in Figure 1 (See Appendix A).

Compound **7**, a white powder soluble in MeOH, has the molecular formula of C_9_H_11_NO_3_ with four degrees of unsaturation from the protonated molecular ion at *m/z* 182.0810 [M + H]^+^ (calculated for C_9_H_12_NO_3_^+^, 182.0812), as evidenced by HR-ESI-MS; the combination of ^13^C-NMR and ^1^H-NMR spectra showed that the compound contained six low-field carbon signals, including four substituted low-field carbons, two unsubstituted aromatic carbons, one submethyl carbon, and two methyl carbons, including one methyl carbon directly connected to the aromatic ring. The compound is an alkaloid, as verified by the bismuth potassium iodide reaction. It is inferred that the compound contained a nitrogen-containing heterocyclic ring. According to the HMBC signal (Table 2), H-5 (*δ*_H_ 7.67) was correlated with C-6, C-3, and C-9, and H-3 was strongly correlated with C-2, C-5, and C-9. Furthermore, the HMBC-related signal intensity was weakly correlated. Combined with hydrogen spectrum signal splitting, C-3 and C-5 were interpositionally substituted, and H-3 and H-5 were weakly correlated with the C-9 signal intensity. H-5 was correlated with C-6 and C-10, and H-10 was strongly correlated with C-6, indicating that C-10 was linked to C-6, and the chemical shift of C-6 was lower than that of conventional aromatic carbonization. It is inferred that C-6 was linked to heteroatom N, resulting in a low chemical shift field, and H-3 and C-2 (*δ*_C_ 166.9) were detected. The correlation signal of H-7 with C-3 and C-8 and the correlation hydrogen spectrum signal of H-8 split (d) indicated that C-2 was connected with a hydroxyethyl, and the abnormal chemical shift of C-2 indicates that it was connected to N. After assignment of the related signals, it was found that C-4 (*δ*_C_ 142.2) did not generate any related signals, and there were two oxygens in the molecular formula of the compound that had not been attributed. Thus, it was inferred that the compound contained carboxylic acid groups. Since the unsaturation degree of the compound was 4 and it was a nitrogen-containing alkaloid, the carboxylic acid groups should be connected with the pyridine ring (Figure 5). The position of C-9 was determined according to the signal correlation between H-3, H-5, and C-9, and the paraposition substitution of the carbonyl group and paraposition N led to the chemical shift of C-4 moving to the lower field: ^1^H-NMR (400 MHz, CD_3_OD) *δ*_H_ 7.89 (br s, 1H, H-3), 7.67 (br s, 1H, H-5), 4.89 (1H, overlapped, H-7), 2.59 (s, 3H, H-10) and 1.46 (d, *J* = 6.6, H-8); ^13^C-NMR (100 MHz; CD_3_OD) *δ*_C_ 168.4 (C-9), 166.9 (C-1), 159.5 (C-6), 142.2 (C-4), 122.6 (C-5), 117.5 (C-3), 71.1 (C-3), 24.4 (C-8), and 23.7 (C-10).

To confirm the stereochemical assignments of **7**, we carried out the ECD calculation at the PBE0/def2-TZVP level. The experimental ECD spectrum of **7** exhibited a negative Cotton effect at 258 nm (Δε −10.54) and a positive Cotton effect at 286 nm (Δε +7.76), which displayed strong agreement with the calculated ECD curve of *S*-**7** (Figure 6). Thus, the absolute configuration at the stereogenic center in **7** was (7*S*). The structure of compound **7** was determined to be 2-(1-hydroxyethyl)-6-methylisonicotinic acid.

Compound **9** was isolated as a light yellow powder, and the HR-ESI-MS data showed a molecule peak at *m/z* 253.0655 [M + H]^+^ (calculated for C_1__2_H_1__3_O_6_^+^, 253.0707), which indicated the molecular formula as C_12_H_12_O_6_ and 7 as the index of hydrogen deficiency; ^1^H-NMR (400 MHz; DMSO-*d*_6_) *δ*_H_ 1.12–1.14 (d, *J* = 6 Hz, 3H, H-3′), 3.77–3.85 (m, *J* = 6.2 Hz, 1H, H-2′), 3.97–4.00 (t, *J* = 5.6 Hz, 1H, H-3′), 3.75–3.77 (d, 1H, OH), 5.64–5.55 (d, 1H, OH), 6.34 (d, *J* = 1.2 Hz, 1H, H-5), 6.44 (d, *J* = 1.2 Hz, 1H, H-7), 6.61 (s, 1H, H-4), 10.85 (s, 1H, H-6), 11.00 (s, 1H, H-8); and ^13^C-NMR (400 MHz; DMSO-*d*_6_) *δ*_C_ 165.5(C-1), 165.3 (C-6), 162.5 (C-8), 157.6 (C-3), 139.2 (C-4a), 104.4 (C-4), 102.9 (C-5), 101.5 (C-7), 97.8 (C-8a), 74.4 (C-1′), 67.4 (C-2′), and 19.7 (C-3′) (Figure 7). The ^1^H-NMR showed that the compound contained one methyl group, two methoxymethyl groups, two alcohol hydroxyl groups, one phenolic hydroxyl group, one phenolic hydroxyl group forming an intramolecular hydrogen bond, and two double-bond protons. The ^13^C-NMR spectrum (DMSO-*d*_6_) of the compound showed 12 carbon signals, including five oxygen-linked aromatic carbons, four oxygen-free aromatic carbons, two oxygen-linked methylene carbons, and one methyl carbon. The unsaturation of the compound was calculated as 6 according to the molecular formula of the compound, and it was inferred that the compound contained two amphioxic heterocyclic systems. H-7 (*δ* 6.33) was correlated with C-5 (*δ* 102.9), C-8a (*δ* 97.8), C-8 (*δ* 162.5), and C-6 (*δ* 165.3), and H-5 was correlated with C-4 (*δ* 104.4), C-6 (*δ* 165.3), and C-8a (*δ* 97.8). Among them, oxygen substitution existed in C-6 (*δ* 165.3), C-8 (*δ* 162.5), and C-1 (*δ* 165.54). According to a comparison of the hydrogen spectra of CD_3_OD, C-6 and C-8 are hydroxyl substitutions, and C-1 is the carbonyl group. The abnormal chemical shift of the C-8 hydroxyl group (*δ* 11.00) indicated that it was greatly shifted to the lower field under the influence of the neighboring carbonyl group. The low-field shift for C-3 (*δ* 157.6) and the HSQC signal suggested the existence of substitution. On the basis of a combination with the HMBC and chemical shift characteristics of C-3 and C-1, we inferred that the two carbons were connected by oxygen, resulting in a large chemical shift to the low field (Table 3). On the basis of the above information, we inferred that the compound was a derivative of an isocoumarin skeleton. There was a correlation between C-4, C-3, and C-1, and it was inferred that C-3 had a branched chain substitution: H-1′, C-2′, C-3′, and H-3′ were correlated with C-1′ and C-2′. A combination of the H spectrum splitting characteristics and hydrogen integral values of H-1, H-2, and H-3 (dd, dq, d peaks; 1H, 1H, 3H, respectively) and the chemical shifts of C-1′, C-2′, and C-3′ (74.4, 67.4, and 19.7, respectively) determined that C-1′ and C-2′ were hydroxyl substituted and C-3′ was a methyl, and the compound signal was assigned. The absolute configurations of compound **9** were established to be 1′*R* and 2′*R* by the ECD calculations (Figure 8). Finally, compound **9** was named 6,8-dihydroxy-3-(1′*R*, 2′*R*-dihydroxypropyl)-isocoumarin (Figure 1).

In addition to the new compounds described above, eight known compounds obtained in this study were identified as compounds **1**–**3**, **5**–**6**, **8**, and **10**–**11** by comparing their spectroscopic data to those reported in the literature. Details of NMR and MS data for compounds **1**–**11** were given in the Appendix A.

Compound **1** was obtained as a colorless powder, and the HR-ESI-MS data showed a molecule peak at *m/z* 129.0552 [M + H]^+^ (calculated for C_6_H_9_O_3_^+^, 129.0546), which indicated a molecular formula of C_6_H_8_O_3_ with three degrees of unsaturation. In examining the proton nuclear magnetic resonance (^1^H-NMR) data, we found signals for methyl protons *δ*_H_ 1.41 (3H d, *J* = 6.8 Hz, H-CH_3_), one methylene proton *δ*_H_ 4.28 (2H, d, *J* = 1.7 Hz), and two methines (two oxygenated sp^3^ and one sp^2^): *δ*_H_ 5.11–5.16 (1H, m, H-5) and *δ*_H_ 7.42–7.47 (1H, m, H-4). The ^13^C-NMR spectra revealed six carbon signals: one ester carbonyl (δ_C_ 174.53), one olefinic methine (*δ*_C_ 152.93), one nonprotonated sp^2^ carbon (*δ*_C_ 134.75), two oxygenated methines (*δ*_C_ 80.13), one oxygenated methylene (*δ*_C_ 56.95), and methyls (*δ*_C_ 19.02). The structure of **1** was determined as 3-(hydroxymethyl)-5-methylfuran-2(5H)-one [28], as shown in Figure 1.

Compound **2** was a white powder with the molecular formula (C_7_H_8_O_4_): HR-ESI-MS *m/z* 157.0493 [M + H]^+^ (calculated for C_7_H_9_O_4_^+^, 157.0495); ^1^H-NMR (400 MHz, CD_3_OD) *δ*_H_ 1.33 (d, *J* = 6.8 Hz, 3H, H-CH_3_), 5.05–5.09 (m, 1H, H-5), and 7.41–7.45 (m, 1H, H-4); and ^13^C-NMR (100 MHz, CD_3_OD) *δ*_C_ 18.93 (C-CH_3_), 31.08 (C-2′), 80.00 (C-5), 127.99 (C-3), 155.38 (C-4), 173.10 (C-2), and 175.47 (C-1′). Thus, compound **2** was identified as [2-(5-methyl-2-oxo-2,5-dihydrofuran-3-yl)-acetic acid] by comparing NMR reference data [29]. Previously, compound **2** had only been obtained through chemical synthesis and was isolated as a natural product for the first time [29]. Thus, we named it aspilactonol G.

Compound **3** was a colorless oil with the molecular formula (C_8_H_10_O_4_): HR-ESI-MS (*m/z* 171.0654 [M + H]^+^ (calculated for C_8_H_10_O_4_^+^, 171.0652)). ^1^H-NMR data (400 MHz, CD_3_OD) *δ*_H_ 1.33 (d, *J* = 6.8 Hz, 3H, H-5), 5.05–5.09 (m, 2H, H-2), and 7.41–7.45 (m, 1H, H-4); and ^13^C-NMR (100 MHz, CD_3_OD) *δ*_C_ 18.92, 30.09, 52.68, 80.05, 127.55, 155.58, 171.64, and 175.25. Thus, compound **3** was identified as [methyl-2-(5-methyl-2-oxo-2,5-dihydrofuran -3-yl)-acetate] by comparing NMR reference data [29]. Compound **3** was also obtained as a natural product for the first time [29]; thus, we named it aspilactonol H.

Compound **5** yielded the following data: HR-ESI-MS *m/z* 131.0663 [M + H]^+^ (calculated for C_6_H_11_O_3_^+^, 131.0703); ^1^H-NMR (400 MHz, CD_3_OD) *δ*_H_ 2.11 (d, 3H, C-6), 2.30–2.34 (t, 2H, H-4), 3.67–3.70 (t, 2H, H-5), and 5.76 (s, 1H, H-2); and ^13^C-NMR (400 MHz, CD_3_OD) *δ*_C_ 18.5, 44.6, 60.9, 120.4, 152.7, and 172.1. Compound **5** is E-Δ^2^-anhydromevalonic acid [30].

Compound **6** yielded the following data: HR-ESI-MS *m/z* 113.0604 [M + H]^+^ (calculated for C_6_H_9_O_2_^+^, 113.0597); 1H-NMR (400 MHz, CD_3_OD), *δ*_H_ 5.79 (1H, q, *J* = 1.5 Hz, H-3), 4.38 (2H, t, *J* = 6.0 Hz, H-6), 2.41 (2H, br.t, *J* = 6.0 Hz, H-5), and 2.02 (3H, s, H-7); and ^13^C-NMR (100 MHz, CD_3_OD) *δ*_C_ 22.6, 28.8, 65.6, 116.2, 158.0, and 164.4. Compound **6** is 4-methyl-5,6-dihydropyren-2-one [31].

Compound **8** was a white amorphous powder. The molecular formula, C_10_H_10_O_4_, was determined by HR-ESI-MS 195.0651 *m/z* [M + H]^+^ (calculated for C_10_H_11_O_4_^+^, 195.0652) and ^13^C-NMR data, corresponding to six degrees of unsaturation; ^1^H-NMR (400 MHz, CD_3_OD) *δ*_H_ 6. 21 (1H, s, H-5), 6.20 (1H, s, H-7), 4.63–4.67 (1H, m, H-3), 2.92 (1H, dd, *J* = 16.4, 3.6 Hz, H-4a), 2.82 (H, dd, *J* = 16.4, 11.2 Hz, H-4b), and 1.45 (3H, d, *J* = 6.3 Hz, H-9); and ^13^C-NMR (CD_3_OD, 100 MHz) *δ*_C_ 170.30 (C-1), 164.99 (C-8), 164.23 (C-6), 142.08 (C-4a), 106.53 (C-5), 100.82 (C-8a), 100.0 (C-7), 75.77 (C-3), 34.13 (C-4), and 19.44 (C-9). Compound **8** was identified as (*R*)-6-hydroxymellein [32].

Compound **10**, C_11_H_10_O_4_, was a yellow powder: HR-ESI-MS *m/z* 207.0651 [M + H]^+^ (calculated for C_11_H_11_O_4_^+^, 207.0652); ^1^H-NMR (400 MHz, CD_3_OD, *δ*, ppm) *δ*_H_ 6.39 (^1^H, d, *J* = 1.5 Hz, H-7), 6.29 (1H, d, *J* = 1.5 Hz, H-5), 6.24 (1H, s, H-4), 3.78 (3H, s, H-10), and 2.10 (3H, s, H-9); and ^13^C-NMR (100 MHz, CD_3_OD) *δ*_C_ 165.1 (C, C-1), 163.6 123 (C, C-8), 158.3 (C, C-6), 155.0 (C, C-3), 142.2 (C, C-4a), 103.5 (CH, C-4), 102.9 (CH, C-5), 100.4 (C, C-8a), 99.2 (CH, C-7), 56.8 (CH_3_, C-10), and 19.4 (CH_3_, C-9). Compound **10** is 6-hydroxy-8-methoxy-3-methylisocoumarin [33].

Compound **11**, C_12_H_12_O_5_, was a yellow powder: ESI-MS *m/z* 237.0765 [M + H]^+^ (calculated for C_12_H_13_O_5_^+^, 237.0757); ^1^H-NMR (400 MHz, CD_3_OD, δ, ppm), *δ*_H_ 6.40 (2H, d, *J* = 2.1, H-5, 7), 6.37 (1H, d, *J* = 2.1, H-4), 4.69 (1H, m, H-2′), 2.59 (2H, m, H-1′), and 1.26 (3H, d, *J* = 6.2, H-3′); and ^13^C-NMR (100 MHz, CD_3_OD, *δ*, ppm), *δ*_C_ 167.8 (C, C-1), 167.3 (C, C-8), 164.8 (C, C-6), 156.2 (C, C-3), 141.3 (CH, C-4), 107.0 (C, C-7), 103.7 (C, C-10), 102.6 (CH, C-9), 99.8 (CH, C-5), 66.2 (CH, C-2′), 43.8 (CH_2_, C-1′), and 23.3 (CH_3_, C-3′). The absolute configuration of compound **11** was established to be 2′*R* by the ECD calculations (Figure 9). Thus, compound **11** was identified as de-*O*-methyldiaporthin by comparing NMR reference data [34,35].

The biosynthesis of the isolated compounds **1**–**4** and **8**–**11** was proposed as shown in Figure 1 and Figure 2, respectively. Furan ring groups are abundant in natural products and play important roles in the pharmacophore of bioactive substances. Furanone and its derivatives have been shown to inhibit the formation of bacterial biofilms; interfere with bacterial population effects; and have analgesic, anti-inflammatory, anticancer, anticonvulsive, antibacterial, antifungal, antioxidation, and other activities. Most of the furanone compounds are synthesized by a single polyketone pathway, although chain fusion of furanone has also been reported in recent years [19,36]. In this study, compounds **4** and **9** possessed o-diol side chains. Different carbon skeletons with the same o-diol side chains suggested the presence of specific hydroxylating enzymes. A plausible biosynthetic pathway for compounds **1**–**4** is proposed (Figure 1). Furanone **1**–**4** are derivatives of α,β-unsaturated γ-lactone. Their synthesis begins with the condensation of five molecules of acetyl-CoA to form the intermediate **a**, which is reduced to generate the critical intermediate **b**, and forms the intermediate **c** under the action of cyclase. Then, **c** undergoes ring-opening and oxidation to generate the intermediate **d**, which is dehydrated to produce compound **4**. Compound **2** is synthesized from **d** through an undefined pathway, then methylated to compound **3** and decarboxylated to compound **1** [19,36]. Compounds **8**–**11** are isocoumarin derivatives (Figure 2), whose biosynthesis by the polyketone synthesis pathway begins with acetyl-CoA. Isocoumarin derivatives have been detected in both plants and microorganisms. The C-8 of isocoumarins does not have the oxidation found in plants, while the isocoumarins commonly found in microorganisms have oxidation at the C-8 position, which is considered to be the biological source of the two types of isocoumarin [37,38].

The AChE inhibitory activities of the crude extracts were evaluated using Ellman’s method, with Rivastigmine and Hup A as the control groups [22,39]. Ethyl acetate, with an inhibition effect value of 82.68%, exhibited better inhibition against AChE than did either petroleum ether extract (47.23%) or buthanol extract (15.82%) (Appendix A). In addition, all of the compounds were investigated for their anti-AChE activities. Compounds **1**–**3** and **5**–**10** exhibited no inhibitory activity against AChE. Compounds **4** and **11** displayed moderate inhibitory effect on AChE activities with IC_50_ values of 6.26 and 21.18 µM, respectively (Table 4).

Structurally, almost all of the furanone compounds in this study contained an α,β-unsaturated carboxylic acid lactone moiety, which might be the key functional group for their biological activity. De-*O*-methyldiaporthin (**11**) was first reported in 1988 [34,35]. It can be used as a microbial herbicide due to its very strong phytotoxic activity [40]. AChEIs are drugs that can be used clinically to treat or alleviate symptoms of AD. They are primarily associated with the direction and efficacy of AD drug development based on the cholinergic injury hypothesis. Thus far, two generations of five AChEI drugs (Tacrine, Donepezil, Rivastigmine, Galantamine, and HupA) have been successfully developed and have become the first choice for clinical treatment or mitigation of AD [10]. Existing clinical AChEI drugs have limitations such as limited efficacy, significant toxicity, and drug resistance. In this study, furanone compound **4** and isocoumarin compound **11** were found to have the potential to inhibit AChE. To the best of our knowledge, furanone compounds were reported here for the first time for their AChE inhibitory activity. Their mechanisms of action and structure–activity relationships in inhibiting AChE require further study by inhibition kinetics analysis and molecular docking methods.

## 4. Conclusions

To summarize, 11 polyketide derivatives, which included three new compounds, aspilactonol I (**4**), 2-(1-hydroxyethyl)-6-methylisonicotinic acid (**7**), and 6, 8-dihydroxy-3-(1′R, 2′R-dihydroxypropyl)-isocoumarin (**9**), and two new natural-sources-derived aspilactonols (G, H) (**2**, **3**) were isolated from an endophytic fungus *Phaeosphaeria* sp. LF5 of *H. serrata*. Their absolute configurations of three new compounds (**4**, **7**, and **9**) and known compound **11** were determined by ECD calculations. Furanone compound **4** and isocoumarin compound **11** exhibited potent AChE inhibitory activities. This study indicates that *Phaeosphaeria* sp. LF5 from *H. serrata* may contain various AChEI compounds, which is a potential resource pool for bioprospecting and isolating AChEIs. Furthermore, this research also provided a material basis for the development of new and efficient AChEI drugs.

## Data Availability

All data generated or analyzed in this study are available within the manuscript and are available from the corresponding authors upon request.

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
