# Peer review of "Polyketide Derivatives from the Endophytic Fungus *Phaeosphaeria* sp. LF5 Isolated from *Huperzia serrata* and Their Acetylcholinesterase Inhibitory Activities"

_jof, 2022, doi:10.3390/jof8030232_

Round 1

Reviewer 1 Report

This manuscript reports new polyketides from Phaeosphaeria sp. and their AchE inhibitory activity as well. The manuscript is not of sufficient quality to be me moved forward for the following reasons:

  1. The whole submission does not look like a final version of the manuscript. The English language is very poor and the documents seems not to have been written by a natural products Chemist.
  2. The structure elucidation of new compounds is not easy to follow and not convincing. The authors should provide with both experimental and calculated values for the molecular masses. Known compounds are briefly described or reported rather at the end of the section. Compound 4 is a new derivative of compound 1 and this should ease the structure elucidation of compound 4, same for compounds 8 and 9.
  3. Compound 7 is completely different from others. Was this compound detected or found in the crude extract? Please check and provide with the UPLC-MS result.
  4. The Proposed biosynthetic pathways for furanone compounds is not well explained.
  5. The supporting information including the NMR data (1D and 2D) was missing

Author Response

February 21, 2022

Dr. Du Zhu

Professor

Key Laboratory of Protection and Utilization of Subtropic Plant Resources of Jiangxi Province

Jiangxi Normal University, Ziyang road 99, Nanchang 330022, China

                              E-mail: zhudu12@163.com

RE: Manuscript ID: jof-1604364

Dear Ms. Karry Han and Reviewers,

Thank you for your letter and for the reviewers’ comments concerning our manuscript Title: Polyketides derivatives from endophytic fungus Phaeosphaeria sp. LF5 of Huperzia serrata and their acetylcholinesterase inhibitory activities, Authors: Yiwen Xiao, Weizhong Liang, Zhibin Zhang, Ya Wang, Shanshan Zhang, Jiantao Liu, Jun Chang, Changjiu Ji, Du Zhu * (Manuscript ID: jof-1604364). Those comments are all valuable and very helpful for revising and improving our paper, as well as the important guiding significance to our researches. We have carefully improved and revised according to the reviewers’ suggestions. And the discussion has be revised as your suggestion. All changes made in the revised manuscript were highlighted in red color in the revised manuscript.

Best regards,

Dr. Du Zhu

Professor

Jiangxi Normal University

Manuscript ID: jof-1604364

Title: Polyketides derivatives from endophytic fungus Phaeosphaeria sp. LF5 of Huperzia serrata and their acetylcholinesterase inhibitory activities

Authors: Yiwen Xiao, Weizhong Liang, Zhibin Zhang, Ya Wang, Shanshan Zhang,

Jiantao Liu, Jun Chang, Changjiu Ji, Du Zhu *

Point-by-point response to Reviewer 1 Comments

This manuscript reports new polyketides from Phaeosphaeria sp. and their AchE inhibitory activity as well. The manuscript is not of sufficient quality to be me moved forward for the following reasons: 

  1. The whole submission does not look like a final version of the manuscript. The English languageis very poor and the documents seems not to have been written by a natural products Chemist.

Response: Thank you very much for your valuable comments, which help me a lot to improve this manuscript. In the newly revised version, we have tried our best to polish the language in our manuscript and doubled check the gramma for every sentence.We have replenished the NMR data (1D and 2D) in the supporting information. In the meanwhile, we also have made some changes in the revised version. All changes in the newly revised version were highlighted with red color in the revised manuscript.

In addition, the manuscript has carefully been proofread by the English language editing service of MDPI, and the attached is the Certificate of English Editing.

  1. The structure elucidation of new compounds is not easy to follow and not convincing. The authors shouldprovide with both experimental and calculated values for the molecular masses. Known compounds are briefly described or reported rather at the end of the section. Compound 4 is a new derivative of compound 1 and this should ease the structure elucidation of compound 4, same for compounds 8 and 9.

Response: Thank you for your constructive comments. Since our previous description might not be quite clear, we have made some changes in the revised version. And we have provided with both experimental and calculated values for the molecular masses as your suggestion. Moreover, the briefly description of known compounds is put at the end of the section. All the detailed revisions/changes in the newly revised version are highlighted with red color in the revised manuscript.

  1. Compound 7 is completely different from others. Was this compound detected or found in the crude extract? Please check and provide with the UPLC-MS

Response: Thank you very much for your valuable comments. Yes, compound 7 can be found in the EA crude extract. In addition, we also detected derivatives of compound 7. Unfortunately, their concentrations were too low to be isolated. The UPLC-MS results are as follows:

Figure R1 UPLC-MS analysis of EA extract (10%-100% CH3OH gradient elution 0.3 mL/min)

Figure R2 HPLC chromatogram of Subfraction Fr. 7.1 with the moving phase of H2O:CH3OH=95:5, 1 mL/min.

  1. The Proposed biosynthetic pathways for furanone compounds is not well explained.

Response: Thank you for your comments. Our previous description might not be quite clear, we have made some changes in the revised version. All changes in the newly revised version were highlighted. The detailed revisions/changes are as follows: “A plausible biosynthetic pathway for compounds 1-4 is proposed (Scheme 1). Furanones 1-4 are derivatives of α, β-unsaturated γ-lactone. Their synthesis begins with the condensation of five molecules of acetyl-CoA to form the intermediate a, which is reduced to generate the critical intermediate b, and forms the intermediate c under the action of cyclase. Then, c undergoes ring-opening and oxidation to generate the intermediate d, which is dehydrated to produce compound 4. Compound 2 is synthesized from d through an undefined pathway, then methylated to compound 3 and decarboxylated to compound 1 [19, 36].”

  1. The supporting informationincluding the NMR data (1D and 2D) was missing

Response: Thank you for your comments. We apologized for the mistake. In the newly revised version, we have replenished the NMR data (1D and 2D) in the supporting information.

Reviewer 2 Report

review for

Polyketides derivatives from endophytic fungus Phaeosphaeria  sp. LF5 of Huperzia serrata and their acetylcholinesterase  inhibitory activities

authors have full legitimacy on this subject as they previously published

Wang, Y.; Lai, Z.; Li, X.X.; Yan R.M.; Zhang Z.B.; Yang, H.L.; Zhu D. Isolation, diversity and acetylcholinesterase inhibitory  activity of the culturable endophytic fungi harboured in Huperzia serrata from Jinggang Mountain, China. World J Microbiol  Biotechnol 2016, 32, 20. DOI:10.1007/s11274-015-1966-3.

Huperzia serrata

Huperzia serrata

Conservation status

Secure (NatureServe)

Scientific classification

Kingdom:

Plantae

Clade:

Tracheophytes

Clade:

Lycophytes

Class:

Lycopodiopsida

Order:

Lycopodiales

Family:

Lycopodiaceae

Genus:

Huperzia

Species:

H. serrata

Binomial name

Huperzia serrata

(Thunb. ex Murray) Trevis.

Huperzia serrata, the toothed clubmoss,[1] is a plant known as a firmoss which contains the acetylcholinesterase inhibitor huperzine A.[2] The species is native to eastern Asia (China, Tibet, Japan, the Korean peninsula, the Russian Far East).[3]

very interesting plant

Huperzia serrata is a Lycophyllaceae plant called as “Shezucao” 46 in China [10].

47 Huperzine A (HupA) was first isolated from H. serrata in the 1980s, and approved in the

48 1990s in China as an acetylcholinesterase inhibitor (AChEI) to treat Alzheimer’s disease

49 (AD). The exiting studies showed that endophytic fungi of H. serrata could not only

50 synthesize huperzine A (HupA) or its similar compounds as host plants, but also contain

51 a lot of novel compounds

Question: what is the current exact picture about the true origin of huperzine A (HupA)?

lines 49-50  H.serrata could not only synthesize huperzine A (HupA) or its similar compounds as host plants

fungi only? plant and fungi?

To summarize, eleven polyketides derivatives including three new compounds

453 aspilactonol I (4), 2-(1-hydroxyethyl)-6-methylisonicotinic acid (7) and 6,

454 8-dihydroxy-3-(1 ′ R, 2 ′ R-dihydroxypropyl)-isocoumarin (9), two new natural

455 sources-derived aspilactonols (G, H) (2, 3), were isolated from an endophytic fungus

456 Phaeosphaeria sp. LF5 of H. serrata. Their absolute configurations were determined by

457 ECD calculations. Furanone compound 4 and isocoumarin compound 11 exhibited

458 potent AChE inhibitory activities.

This paper is mainly focused on chemistry of fungal products, not on product level optimization, culture conditions etc  (fully in line with JoF editorial policy?)

In conclusion, this is a nice study, very helpful, as it indicates that Phaeosphaeria sp. LF5 from H. serrata may contain various AChEI compounds, which is a potential resource pool for bioprospecting and isolating AChEIs. Furthermore, this research also provided a material basis for the development of new and efficient AChEI drugs.

Author Response

February 21, 2022

Dr. Du Zhu

Professor

Key Laboratory of Protection and Utilization of Subtropic Plant Resources of Jiangxi Province

Jiangxi Normal University, Ziyang road 99, Nanchang 330022, China

                              E-mail: zhudu12@163.com

RE: Manuscript ID: jof-1604364

Dear Ms. Karry Han and Reviewers,

Thank you for your letter and for the reviewers’ comments concerning our manuscript Title: Polyketides derivatives from endophytic fungus Phaeosphaeria sp. LF5 of Huperzia serrata and their acetylcholinesterase inhibitory activities, Authors: Yiwen Xiao, Weizhong Liang, Zhibin Zhang, Ya Wang, Shanshan Zhang, Jiantao Liu, Jun Chang, Changjiu Ji, Du Zhu * (Manuscript ID: jof-1604364). Those comments are all valuable and very helpful for revising and improving our paper, as well as the important guiding significance to our researches. We have carefully improved and revised according to the reviewers’ suggestions. And the discussion has be revised as your suggestion. All changes made in the revised manuscript were highlighted in red color in the revised manuscript.

Best regards,

Dr. Du Zhu

Professor

Jiangxi Normal University

Manuscript ID: jof-1604364

Title: Polyketides derivatives from endophytic fungus Phaeosphaeria sp. LF5 of Huperzia serrata and their acetylcholinesterase inhibitory activities

Authors: Yiwen Xiao, Weizhong Liang, Zhibin Zhang, Ya Wang, Shanshan Zhang,

Jiantao Liu, Jun Chang, Changjiu Ji, Du Zhu *

Point-by-point response to Reviewer 2 Comments

review for

Polyketides derivatives from endophytic fungus Phaeosphaeria sp. LF5 of Huperzia serrata and their acetylcholinesterase inhibitory activities

authors have full legitimacy on this subject as they previously published

Wang, Y.; Lai, Z.; Li, X.X.; Yan R.M.; Zhang Z.B.; Yang, H.L.; Zhu D. Isolation, diversity and acetylcholinesterase inhibitory activity of the culturable endophytic fungi harboured in Huperzia serrata from Jinggang Mountain, China. World J Microbiol Biotechnol 201632, 20. DOI:10.1007/s11274-015-1966-3.

Huperzia serrata

Huperzia serrata

Conservation status

Secure (NatureServe)

Scientific classification

Kingdom:

Plantae

Clade:

Tracheophytes

Clade:

Lycophytes

Class:

Lycopodiopsida

Order:

Lycopodiales

Family:

Lycopodiaceae

Genus:

Huperzia

Species:

H. serrata

Binomial name

Huperzia serrata

(Thunb. ex Murray) Trevis.

Huperzia serrata, the toothed clubmoss,[1] is a plant known as a firmoss which contains the acetylcholinesterase inhibitor huperzine A.[2] The species is native to eastern Asia (China, Tibet, Japan, the Korean peninsula, the Russian Far East).[3]

very interesting plant

Huperzia serrata is a Lycophyllaceae plant called as “Shezucao” 46 in China [10].

47 Huperzine A (HupA) was first isolated from H. serrata in the 1980s, and approved in the

48 1990s in China as an acetylcholinesterase inhibitor (AChEI) to treat Alzheimer’s disease

49 (AD). The exiting studies showed that endophytic fungi of H. serrata could not only

50 synthesize huperzine A (HupA) or its similar compounds as host plants, but also contain

51 a lot of novel compounds

Question: what is the current exact picture about the true origin of huperzine A (HupA)?

lines 49-50 Hserrata could not only synthesize huperzine A (HupA) or its similar compounds as host plants

fungi only? plant and fungi?

Response: Thank you very much. This is a good question. The origin of HupA has always been a controversial issue, attracting the attention of many researchers. HupA was first isolated from H. serrata in 1986, the biosynthetic gene cluster of HupA was also found in H. serrata. Meanwhile, HupA has also been isolated from endophytic fungi, such as Penicillium sp. LDL4.4, Colletotrichum gloeosporioides ESO26 and Cg01, Alternaria brassicae AGF041, Penicillium polonicum hy4, Shiraia sp. Slf14, Paecilomyces tenuis YS-13, Trichoderma harzianum L44, Colletotrichum cladosporioides LF70 and Ceriporia lacerate HS-ZJUT-C13A. Because of the important role of endophytic fungi for plant survival (regulating photosynthesis, increasing nutrient uptake, and mitigating the effects of various stresses), they have been selected for co-evolution with their hosts during evolution. The existing studies suggest that genes involved in HupA biosynthesis may be translocated into endophytes with symbiotic and evolutionary adaptations.

[1] Cao, D.; Sun, P.; Bhowmick, S.; Wei, Y.H.; Guo, B.; Wei, Y.H.; Mur, L.A.J.; Sun, Z.L. Secondary metabolites of endophytic fungi isolated from Huperzia serrata. Fitoterapia. 2021, 155, 104970. DOI:10.1016/j.fitote.2021.104970.

[2] Yang, M.; You, W.; Wu, S.; Fan, Z.; Xu, B.; Zhu, M.; Li, X.; Xiao, Y. Global transcriptome analysis of Huperzia serrata and identification of critical genes involved in the biosynthesis of huperzine A. BMC Genomics 2017, 18, 245. https://doi.org/10.1186/s12864-017-3615-8.

[3] Lu, Z.; Ma, Y.; Xiao, L.; Yang, H.; Zhu, D. Diversity of endophytic fungi in Huperzia serrata and their acetylcholinesterase inhibitory activity. Sustainability 202113, 12073. https://doi.org/10.3390/su132112073.

[4] Yang, H.L.; Ma, Y.S.; Wang, X.L.; Zhu, D. Huperzine A: a mini-review of biological characteristics, natural sources, synthetic origins, and future prospects. Russ J Org Chem 2020, 56, 148-157. https://doi.org/10.1134/S1070428020010236.

To summarize, eleven polyketides derivatives including three new compounds

453 aspilactonol I (4), 2-(1-hydroxyethyl)-6-methylisonicotinic acid (7) and 6,

454 8-dihydroxy-3-(1 ′ R, 2 ′ R-dihydroxypropyl)-isocoumarin (9), two new natural

455 sources-derived aspilactonols (G, H) (23), were isolated from an endophytic fungus

456 Phaeosphaeria sp. LF5 of H. serrata. Their absolute configurations were determined by

457 ECD calculations. Furanone compound 4 and isocoumarin compound 11 exhibited

458 potent AChE inhibitory activities.

This paper is mainly focused on chemistry of fungal products, not on product level optimization, culture conditions etc (fully in line with JoF editorial policy?)

Response: Thank you for your valuable comments. Our manuscript is submitted to the special issue of Journal of Fungi (ISSN 2309-608X). This special issue belongs to the section "Fungal Cell Biology, Metabolism and Physiology". The special issue entitled “Secondary Metabolites from Fungi in honor of Prof. Dr. Ji-Kai Liu on the occasion of his 60th birthday”. This special issue is dedicated to all aspects of fungal natural product chemistry, including fungal natural product isolation, structural elucidation, bioactivity, synthesis, biosynthesis, etc.

Two similar articles were published in this special issue of Journal of Fungi, as following:

[1] Yu, J.-J.; Jin, Y.-X.; Huang, S.-S.; He, J. Sesquiterpenoids and Xanthones from the Kiwifruit-Associated Fungus Bipolaris sp. and Their Anti-Pathogenic Microorganism Activity. J. Fungi 20228, 9. https://doi.org/10.3390/jof8010009;

[2] Dai, Q.; Zhang, F.-L.; Li, Z.-H.; He, J.; Feng, T. Immunosuppressive Sesquiterpenoids from the Edible Mushroom Craterellus odoratusJ. Fungi 20217, 1052. https://doi.org/10.3390/jof7121052.

In conclusion, this is a nice study, very helpful, as it indicates that Phaeosphaeria sp. LF5 from H. serrata may contain various AChEI compounds, which is a potential resource pool for bioprospecting and isolating AChEIs. Furthermore, this research also provided a material basis for the development of new and efficient AChEI drugs.

Response: Thank you for your valuable comments. We have carefully improved and revised according to the reviewers’ suggestions. All changes made in the revised manuscript were highlighted in red color in the revised manuscript.

In addition, the manuscript has carefully been proofread by the English language editing service of MDPI, and the attached is the Certificate of English Editing.

Round 2

Reviewer 1 Report

The manuscript has been revised and improved accordingly.